# Superoxide Dismutase-3 Downregulates Laminin α5 Expression in Tumor Endothelial Cells via the Inhibition of Nuclear Factor Kappa B Signaling

**DOI:** 10.3390/cancers14051226

**Published:** 2022-02-26

**Authors:** Lorena Carmona-Rodríguez, Diego Martínez-Rey, Paula Martín-González, Mónica Franch, Lydia Sorokin, Emilia Mira, Santos Mañes

**Affiliations:** 1Department of Immunology and Oncology, Centro Nacional de Biotecnología (CNB/CSIC), 28049 Madrid, Spain; lcarmona@cnb.csic.es (L.C.-R.); diego.martinez.rey@hotmail.com (D.M.-R.); pm681@cam.ac.uk (P.M.-G.); 2Bioinformatics Unit, Centro Nacional de Biotecnología (CNB/CSIC), 28049 Madrid, Spain; monica.franch@perkinelmer.com; 3Institute of Physiological Chemistry and Pathobiochemistry and Cells in Motion Interfaculty Centre, University of Münster, 48049 Münster, Germany; sorokin@uni-muenster.de

**Keywords:** basement membrane, laminin, immunotherapy, oxidative stress, diapedesis, endothelium, inflammation

## Abstract

**Simple Summary:**

Tumor-infiltrating lymphocytes determine cancer prognosis and the response to immunotherapy. The balance of laminin-α4/laminin-α5 isoforms in the endothelial basement membrane plays an instructive function in regulating the diapedesis of immune cells under inflammatory conditions. Previous studies showed that the extracellular superoxide dismutase (SOD)3 induces laminin-α4, which is associated with improved disease-free survival of colorectal cancer patients. The aim of our study was to determine whether SOD3 also affects the expression of laminin-α5 in the tumor vasculature. The results showed that SOD3 differentially regulates laminin-α4 and laminin-α5 in the tumor endothelium. SOD3 promoted notable transcriptomic changes in tumor-stimulated endothelial cells, including the inhibition of the nuclear factor kappa B (NF-κB) pathway, an inductor of laminin α5 transcription. Therefore, high SOD3 levels in the tumor vasculature shifted the laminin α4/α5 balance towards the laminin-α4^high^/laminin-α5^low^ phenotype, which is permissive for T cell diapedesis into tumors and explains the improved cancer immune surveillance associated to high SOD3 levels.

**Abstract:**

The balance between laminin isoforms containing the α5 or the α4 chain in the endothelial basement membrane determines the site of leukocyte diapedesis under inflammatory conditions. Extracellular superoxide dismutase (SOD3) induces laminin α4 expression in tumor blood vessels, which is associated with enhanced intratumor T cell infiltration in primary human cancers. We show now that SOD3 overexpression in neoplastic and endothelial cells (ECs) reduces laminin α5 in tumor blood vessels. SOD3 represses the laminin α5 gene (*LAMA5*), but *LAMA5* expression is not changed in SOD1-overexpressing cells. Transcriptomic analyses revealed SOD3 overexpression to change the transcription of 1682 genes in ECs, with the canonical and non-canonical NF-κB pathways as the major SOD3 targets. Indeed, SOD3 reduced the transcription of well-known NF-κB target genes as well as NF-κB-driven promoter activity in ECs stimulated with tumor necrosis factor (TNF)-α, an NF-κB signaling inducer. SOD3 inhibited the phosphorylation and degradation of IκBα (nuclear factor of the kappa light polypeptide gene enhancer in B-cells inhibitor alpha), an NF-κB inhibitor. Finally, TNF-α was found to be a transcriptional activator of *LAMA5* but not of *LAMA4*; *LAMA5* induction was prevented by SOD3. In conclusion, SOD3 is a major regulator of laminin balance in the basement membrane of tumor ECs, with potential implications for immune cell infiltration into tumors.

## 1. Introduction

Solid tumors are usually irrigated by an immature and dysfunctional vascular network. The hierarchical branching organization and the features associated with arterioles, capillaries and venules are not seen in a tumor’s vasculature. Rather, the vessels are tortuous and dilated with a chaotic blood flow (sometimes even stationary), resulting in the deficient transport of nutrients and oxygen, the latter leading to hypoxia. However, tumor cells not only adapt and survive under these stressful conditions, but also use this abnormal vascular network to create a shield that protects them from the immune system and therapeutic agents. The conversion of this immature endothelium into a more “normal” vascular network is, therefore, a promising therapeutic strategy for enhancing drug delivery and boosting immunotherapy [1,2,3,4,5].

Major drivers of the abnormal nature of tumor vasculature include inflammation, the composition of the extracellular matrix (ECM), and the accumulation of pro-angiogenic factors such as vascular-endothelial growth factor (VEGF)-A, in the tumor microenvironment (TME) [6]. VEGF-A not only triggers the proliferation, migration and sprouting of endothelial cells (ECs), but also greatly increases tumor vessel permeability by weakening vascular endothelial (VE)-cadherin-mediated EC junctions and interfering with pericyte-EC interactions [7,8]. Many of these proangiogenic factors are induced by inflammatory mediators, such as tumor necrosis factor (TNF)-α, acting on tumor and stromal cells via the nuclear factor kappa B (NF-κB) pathway [9]. NF-κB is a transcription factor assembled through the homo- or heterodimerization of five available subunits via the Rel-homology domain (a domain also involved in DNA binding) [10]. When processed proteolytically, the Rel family members p100 and p105 give rise to the p52 and p50 isoforms, respectively; these do not contain transactivation domains and need to interact with other family members, such as RelA (p65), RelB or c-Rel, to function as transcription factors [11,12]. p52/p50 dimers operate as transcriptional repressors [13].

The activity and nuclear translocation of RelA, RelB and c-Rel is regulated by inhibitors of the κB (IκB)-α and -β family proteins, which bind heterodimers in the cytoplasm. Most of the NF-κB-activating signals converge at the level of the IκB kinases (IKKα and IKKβ), which upon activation phosphorylate IκB inhibitors, priming them for ubiquitylation and proteasomal degradation [14]. This releases the NF-κB dimers, allowing them to enter the nucleus. It is noteworthy that all NF-κB signaling molecules are present in ECs, and their sustained activation in cancer and other pathologies leads to vascular disorganization and a reduction in the barrier function of the EC [11,15,16,17].

A central, but commonly overlooked, structural and functional element of blood vessels is the endothelial basement membrane (BM), a self-assembled layer of highly specialized glycoproteins, produced by ECs and pericytes, that underlies the former and encases the latter [18]. Endothelial BMs are not only structural scaffolds, but also contribute to the mechanical properties of blood vessels and transmit signals to endothelial and perivascular cells that may regulate blood flow and nutrient and oxygen supply [19]. Tumor vessels and endothelial sprouts also have a BM, but these membranes show abnormalities reminiscent of degenerated or regenerating vessels, including the loss of association with ECs, a varying thickness, and multiple layers [20]. A full understanding of the characteristics and molecular composition of the endothelial BM in tumor blood vessels is yet to be gained [21].

BM assembly is dependent on the initial self-assembly of laminins [22], a family of heterotrimeric ECM proteins. Sixteen laminin isoforms exist, the product of different combinations of five α, four β and three γ chains [23]. The biological role of the laminins is largely defined by their α-chain interaction with integrin and non-integrin receptors [24,25]. Laminin α4 and α5 chains combine with laminin β1 and γ1 chains to form laminins 411 and 511, respectively, found in endothelial and smooth muscle BMs [18].

Previous studies have shown a patchy distribution of laminin α4 and α5 along the ‘vascular tree’ with little or no laminin α5 at postcapillary venules where leukocyte extravasation preferentially occurs [26,27,28,29,30]. Curiously, leukocyte adhesion to laminin α5-containing laminins is stronger than to laminin α4-containing isoforms, but their migration across laminin α4-positive BMs is faster [30]. The laminin α5/laminin α4 balance in the endothelial BM is, therefore, important in inflammation and potentially also in immune surveillance, as suggested by work in models of neuroinflammation, allograft rejection, and cancer [30,31,32]. In cancer, a recent study has shown a relationship between vascular normalization and changes in the expression of laminin α4-containing isoforms in the BM of tumor blood vessels, driven by high levels of the superoxide dismutase-3 (SOD3) in the TME [33]. Importantly, these SOD3-induced changes were also associated with the increased infiltration of cytotoxic CD8^+^ T cells and prolonged disease-free survival in patients with colorectal cancer.

SOD3 is an antioxidant metalloenzyme that localizes predominantly to the extracellular space where it associates with the ECM [34]. In fibroblasts and ECs, SOD3 can also be found in the nucleus and cytoplasm [35]. It is expressed in most tissues under physiological conditions, and accounts for up to 70% of extracellular SOD activity in the walls of healthy vessels. However, oncogenesis commonly represses SOD3 in many mouse and human tumors, and may contribute to the growth and progression of malignancies (reviewed in [36]). Analyzing how SOD3 re-expression affects the interaction of stromal and cancer cells in the TME is, therefore, of great interest.

SOD3 protects tissues from oxidative damage and prevents the generation of peroxynitrite from nitric oxide (NO) by catalyzing the dismutation of the superoxide anion. SOD3 activity thus preserves NO levels to act as an intra- and intercellular messenger in the endothelium. Indeed, the normalizing effects of SOD3 on the tumor vasculature are dependent on the NO-mediated stabilization of hypoxia-inducible factor (HIF)-2α, a transcription factor that upregulates VE-cadherin and reduces vascular leakage [37]. SOD3 also enhances the entry of T cells into tumors and improves immunotherapy through a mechanism that involves the induction of laminin α4 in a HIF-2α- and WNT-dependent manner [32]. Whether SOD3 also affects the expression of laminin α5 in tumor blood vessels is unknown. The present work shows that SOD3 triggers the transcription of the gene encoding laminin α4 (*LAMA4*) but represses that of the gene coding for laminin α5 (*LAMA5*) in ECs in vitro and in vivo, thus changing the laminin α4/laminin α5 ratio in tumor blood vessels. SOD3 repression of *LAMA5* is dependent on the inhibition of the NF-κB pathway, here identified as a regulator of *LAMA5* transcription in ECs under inflammatory conditions. The present work also identifies a complex transcriptional program under the control of SOD3 in ECs, responsible for the normalization of the tumor vasculature.

## 2. Materials and Methods

### 2.1. Cell Lines, Transfection and Stimulation

The mouse microvascular 1G11-mock and 1G11-SOD3, and the EG7-mock and EG7-SOD3 mouse thymoma cell lines have been described elsewhere [32,37]. Human dermal microvascular endothelial cells (HDMEC; Cascade Biologics, Portland, OR, USA) were cultured and transduced with recombinant adenoviruses encoding for human SOD3 or β-galatosidase (mock) as described [37]. N202.1A murine mammary cancer cells were provided by Dr. V. Bronte (Verona University, Verona, Italy) and cultured as previously described [38]. 1G11-SOD1 cells were generated by retroviral transduction with pRV-SOD1-IRES-GFP, cloned from pF151 pcDNA3.1(+) SOD1WT (a gift from Dr. E. Fisher; Addgene plasmid #26397) [39] (Addgene; Watertown, MA, USA). Transduced cells were selected by cell sorting via GFP fluorescent emission using an MoFlo XDP device (Beckman Coulter; Brea, CA, USA).

HA-tagged SOD3 (pCMVSPORT6-HASOD3) was generated by inserting the HA tag sequence between the signal peptide and the mature SOD3 sequence in pCMVSPORT6-SOD3 (Open Biosystems; Huntsville, AL, USA). HEK-293T cells (ATCC; Manassas, VA, USA) were transfected with pCMVSPORT6-HASOD3 or pCMVSPORT6-SOD3 using Lipofectamine 2000 (Invitrogen; Waltham, MA, USA); 24 h post-transfection, the cells were cultured in basal medium (DMEM plus 0.1% BSA) for three days and cell-free conditioned medium collected and stored at −80 °C. Endothelial origin of 1G11 cells was verified regularly by VE-cadherin staining. Cell-free conditioned medium from N202.1A cells was obtained after cell culture in basal medium for five days. All cell lines were periodically tested for mycoplasmas.

For NF-κB activation, 1G11-mock or 1G11-SOD3 cells were starved for 1 h before stimulation with 1–10 ng/mL recombinant TNF-α (Peprotech; Cranbury, NJ, USA). In some experiments, cells were pretreated with BAY 11-7082 (1–2 µM, 1–2 h, 37 °C) (Selleckchem; Houston, TX, USA), followed by recovery in basal medium (6 h) before TNF-α stimulation.

### 2.2. Tumor Samples 

C57BL/6J mice were supplied by The Jackson Laboratory (Bar Harbor, ME, USA) and SOD3EC-Tg mice generated as previously described [32]. For tumor generation, 2–5 month-old female mice were used. EG7-mock or EG7-SOD3 cells (5 × 10^6^/100 μL) were implanted subcutaneously into the indicated mouse strains (*n* = 7–9 mice/group) and tumors excised after 21 days. All mouse experiments were approved by the Comunidad de Madrid (PROEX 399/15) and the CNB Ethics Committees in strict accordance with the Spanish and European Union laws and regulations concerning the care and use of laboratory animals.

### 2.3. RT-qPCR

Total RNA was extracted from mouse tumors and cell lines using TRI-Reagent (Sigma Aldrich; Burlington, MA, USA). cDNA strands were synthesized from 0.5–2 µg total RNA using the High-Capacity cDNA Reverse Transcription Kit (Applied Biosystems; Waltham, MA, USA), employing random primers. mRNA levels were quantified by qPCR in a Quant-Studio 5 Real-Time PCR System (Thermo Fisher; Waltham, MA, USA) using the SYBR EvaGreen-based reaction mix (5× PyroTaq EvaGreen qPCR Mix Plus ROX) (Cmb-Bioline; Madrid, Spain) and specific primers (Appendix A), employing β-actin for normalization. Relative quantities (Rq) (i.e., relative to the sample with the lowest expression or the control sample) were calculated using the 2^−ΔΔCt^ method.

### 2.4. Microarray Analysis

1G11-mock and 1G11-SOD3 cells were incubated with N202.1A-conditioned medium for 6 h and total RNA isolated with TRI-Reagent. This was then further cleaned using the RNeasy Kit (Qiagen; Hilden, Germany). RNA quality was confirmed by electropherogram, employing an Agilent Technologies 2100 Bioanalyzer (Agilent Technologies; Santa Clara, CA, USA). The preparation of probes and hybridization were performed as described in the One-Color Microarray-Based Gene Expression Analysis Manual Ver. 6.5 (Agilent Technologies). RNA quality check, amplification, labeling, hybridization with Array SurePrint Mouse G3 8 × 60 (Agilent Technologies, G4852A) and initial data extraction were performed at the Genomic Service Facility at the Centro Nacional de Biotecnología (CNB-CSIC). Images were captured with an Agilent Microarray Scanner and spots were quantified using Feature Extraction Software (Agilent Technologies). Correction for the local background was performed using the normexp method with an offset of 50. Background corrected intensities were Log2-transformed and normalized by adjusting their quantiles [40]. Changes in expression over replicates was estimated using an empirical Bayes moderated t statistic (LIMMA) [41]. To control the FDR, probability values were corrected using the Benjamini–Hochberg method [42]. The FIESTA viewer (http://bioinfogp.cnb.csic.es/tools/FIESTA, accessed on 25 February 2022), developed by J.C. Oliveros (CNB Bioinformatics Core Facility), was used to visualize the results. Principal components analysis (PCA) was performed to demonstrate similarity between expression profiles using ClustVis software [43]. A volcano plot was prepared using Prism 7.0 software (GraphPad; San Diego, CA, USA), with differentially expressed genes displayed in red (Fc > 2 or < −2, and FDR LIMMA < 0.05). All data files and normalized values were deposited in the NCBI Gene Expression Omnibus database (http://www.ncbi.nlm.nih.gov/geo, accessed on 25 February 2022).

### 2.5. IPA and GSEA Analyses

The list of differentially expressed genes was subjected to functional analysis using Ingenuity Pathway Analysis (IPA) software v.6875226 (Ingenuity^®^ Systems; Redwood City, CA, USA; www.ingenuity.com, accessed on 25 February 2022). Each gene identifier was mapped to its corresponding gene object in the Ingenuity Pathway Knowledge Base (IPKB). The core analysis was run using the default IPA settings: (i) the data source included all species, tissues and cell lines; (ii) interactions were queried for all genes stored within the IPAKB; and (iii) no fold-change cut-off was specified. The significance of the canonical pathways was calculated according to the IPA user’s manual: (1) the ratio of the number of genes from the dataset that map to the pathway over the total number of genes that make up the pathway was determined; and (2) Fisher’s exact test was used to calculate a *p* value determining the probability that the association between the genes in the dataset and the canonical pathway was explained by chance alone. A pathway was considered significant when *p* < 0.05. 

For Gene Set Enrichment Analysis (GSEA) [44], Log2 ratios and the weight punctuation system were used as classification metrics. The gene sets for the canonical and non-canonical NF-κB pathways were obtained from the Molecular Signatures Database v. 6.0 (http://software.broadinstitute.org/gsea/msigdb/index/.jsp, accessed on 25 February 2022), and the analysis performed with those genes differentially expressed (as determined in microarray analyses) and showing an FDR of <0.25. GSEA was undertaken using the default parameters, that is, collapsing each probe set into a single vector, and using a permutation number of 1000 and the permutation type “gene-sets”.

### 2.6. Immunofluorescence Staining

Mock-, SOD3- or SOD1-transfected 1G11 cells were seeded on 1% gelatin-coated Nunc Lab-Tek chamber slides (Nunc; Rochester, NY, USA), fixed with methanol (10 min, −20 °C), and stained with rabbit anti-laminin α5 (antibody 504, described in [45]; or PAC078Mu01 antibody, Cloud Clone; Houston, TX, USA) or goat anti-laminin α4 (N20, sc-16589) (Santa Cruz Biotechnology; Dallas, TX, USA), followed by Alexa488- or Alexa594-labeled secondary antibodies (Thermo Fisher). The staining of 1G11 cells with anti-hemagglutinin (HA) tag antibody (clone 16B12) (Covance; Burlington, NC, USA) and p65 (RelA, clone L8F6) (Cell Signaling; Danvers, MA, USA) was performed after fixation with 4% paraformaldehyde (15 min, 20 °C) and permeabilization with 0.3% Triton X-100 (5 min, 20 °C). Tumor sections (10 µm) from snap-frozen (in OCT medium) (Sakura; Torrance, CA, USA) samples were fixed with cold acetone and used for immunostaining with anti-CD31 (MEC13.3) (BD Biosciences Pharmingen; San Diego, CA, USA), rabbit anti- laminin α5 (antibody 504) or goat anti-laminin α4 (N20, sc-16589; Santa Cruz Biotechnology), followed by appropriate fluorescently labeled secondary antibodies. In all cases, samples were mounted in 4,6-diamidino-2-phenylindole (DAPI)-containing Fluoromount-G (SouthernBiotech; Birmingham, AL, USA). Images were analyzed using a Leica DM RB (Wetzlar, Germany) equipped with an Olympus DP70 camera (Tokyo, Japan) at 200× or 400× total magnification (20× or 40× objective), or an Olympus FluoView 1000 confocal microscope with a 60× 1.4 oil plan-Apo objective. 

For quantification of laminin α5 and laminin α4 staining intensities in tumor sections, images were transformed to 8-bit versions with ImageJ software and, after threshold adjustment, the Image calculator tool was used to select coincident CD31- and laminin α5/α4-stained regions. The laminin/CD31 ratio was then calculated. For laminin α5 and laminin α4 quantification in cultured ECs, mean fluorescence intensity was quantified using the ImageJ tool. An in-house open-source MATLAB-based computational platform was developed and used for automatic quantification of p65 nuclear staining. Nuclei were segmented using DAPI images and this segmentation used with the Alexa546 channel to measure the p65-stained pixels within the nuclear area; the ratio of nuclear staining to total nuclear area was then calculated.

### 2.7. Immunoblot and Chase Experiments

1G11 transfectants and human HEK-293T (used as a human SOD1 positive control) cells were lysed with RIPA buffer, and the products resolved by SDS-PAGE and transferred to membranes for immunoblot analysis with anti-SOD1 (NBP2-24915) (NOVUS Biologicals; Centennial, CO, USA), anti-p65 (L8F6) (Cell Signaling; Danvers, MA, USA), anti-β-actin (AC-15) (Sigma Aldrich; Burlington, MA, USA) or anti-tubulin (DM 1A) (Sigma Aldrich; Burlington, MA, USA) antibodies. SOD3 or HA-SOD3 was analyzed in conditioned medium using anti-SOD3 (M-106, sc-67089) (Santa Cruz Biotechnology) or anti-HA (clone 16B12) (Covance; Burlington, NC, USA) antibodies, respectively; loading control was by Ponceau staining. For laminin α4 and laminin α5 detection, RIPA cell extracts from 1G11-mock and 1G11-SOD3 cells were resolved in 4%–12% NuPAGE gels in MOPS SDS running buffer (ThermoFisher; Waltham, MA, USA), transferred to membranes using 48 mM Tris-HCl pH 8.3, 390 mM glycine, 0.1% SDS, 20% methanol, and immunoblotted sequentially with anti-laminin α5 (antibody 504), anti-laminin α4 (antibody 377b), anti-SOD3 and anti-β-tubulin (clone DM1A; Sigma Aldrich; Burlington, MA, USA) antibodies.

For chase experiments, 1G11-mock and 1G11-SOD3 cells grown in basal medium were pretreated (1 h, 37 °C) with 100 µg/mL cycloheximide (CHX; Sigma Aldrich; Burlington, MA, USA), followed by TNF-α (1 ng/mL) stimulation still in the presence of CHX. After cell lysis and transfer, membranes were blotted with anti-IκBα (44D4) (Cell Signaling; Danvers, MA, USA) and anti-tubulin antibodies. The IκBα/tubulin ratio was calculated from densitometry using ImageJ software. Phospho-IκBα kinetics were analyzed in 1G11-mock or 1G11-SOD3 cells grown in basal medium after treatment with the proteasome inhibitor MG132 (10 µM) (Selleckchem; Houston, TX, USA). Cell lysates were immunoblotted with anti-phospho(Ser32/36)-IκBα (5A5) (Cell Signaling; Danvers, MA, USA) and anti-tubulin antibodies, and the ratio calculated as above. 

### 2.8. SOD Activity Assays

Exponentially growing 1G11-mock and 1G11-SOD1 cells were incubated (16 h) with N202.1A cell-conditioned medium in the presence or absence of the SOD mimetic Mn(III)tetrakis(4-benzoic acid)porphyrin (MnTBAP; 100 µM) (Calbiochem; San Diego, CA, USA). Cells were then stained with the cell-permeating superoxide probe dihydroethidium (DHE; 2 µM, 30 min, 37 °C) (Invitrogen; Waltham, MA, USA), which becomes red after oxidation. Cell extracts from 1G11-mock and SOD1 cells and conditioned medium from HEK-293T cultures enriched in untagged or HA-tagged SOD3 were assayed for SOD activity using the OxiSelect Superoxide Dismutase Activity Assay Kit (Cell Biolabs; San Diego, CA, USA) according to the manufacturer’s indications. 

### 2.9. NF-κB Promoter Assay

1G11-mock and 1G11-SOD3 cells were co-transfected with p(NF-κB)3x-LUC (kindly provided by Dr. M. Ricote [CNIC]) and pRL-SV40-LUC plasmids (ratio 4:1). After 48 h, cells were stimulated with N202.1A-conditioned medium, and firefly and *Renilla* luciferase activities measured using the Dual Luciferase Reporter Assay System (Promega; Madison, WI, USA).

### 2.10. ELISA and Flow Cytometry

CCL2 was measured in cell-free supernatants collected from 1G11-mock and 1G11-SOD3 cultures (48 h) by ELISA using the mouse MCP-1 ELISA MAX Standard Set (Biolegend; San Diego, CA, USA). For VCAM-1 analysis, 1G11-mock and 1G11-SOD3 cells were stained with anti-CD106-biotin antibody (clone 429) (Biolegend) followed by streptavidin-phycoerythrin, and analyzed by FACS using a FC-500 Cytomics device (Becton Dickinson; Franklin Lakes, NJ, USA).

### 2.11. Bioinformatic Analyses

The association between *LAMA5* levels and CD8^+^ T cell infiltration was analyzed by TIMER algorithm (https://cistrome.shinyapps.io/timer/, accessed on 25 February 2022). Spearman’s correlation coefficients and *p*-values were provided by the program. The relationship between *LAMA5* expression and tumor prognosis was analyzed using the Kaplan-Meier plotter database (http://kmplot.com/analysis/, accessed on 25 February 2022) [46]. 

### 2.12. Statistical Analyses

Data are shown as means ± SEM unless otherwise indicated. Individual values are also shown; the number of replicates is given in figure legends. For data with a Gaussian distribution and homogeneity of variance, differences were examined using the two-tailed Student’s *t*-test for comparison of two independent groups, and one- or two-way ANOVA with Dunnett’s or Bonferroni’s post-hoc correction for multiple comparisons. When these requirements were not fulfilled, data were analyzed using non-parametric tests (Mann-Whitney). Significance was set at *p* < 0.05. All calculations were performed using Prism 7.0 software.

## 3. Results

### 3.1. SOD3 Downregulates Laminin α5 Expression in the Tumor Vasculature

Previous studies in immunocompetent mice with EG-7 tumors (a thymoma) showed that the forced expression of SOD3 in neoplastic and endothelial cells correlated with enhanced perivascular levels of laminin α4-containing isoforms [32]. In addition, SOD3 overexpression in tumor and endothelial cells was associated with an improved response to the adoptive transfer of tumor-specific CD8^+^ T cells, suggesting that SOD3 provides a permissive signal for tumor infiltration by these cells [32]. Here, these tumor samples were analyzed by immunofluorescence to see whether SOD3 also affects laminin α5 expression in the tumor vasculature. 

In tumors induced by control (EG7-mock) and SOD3-overexpressing (EG7-SOD3) cancer cells (Figure 1A), laminin α5 staining only appeared in close proximity to the endothelial cell marker CD31^+^ (Figure 1B), suggesting localization in the endothelial BM. Importantly, SOD3 overexpression in neoplastic cells was associated with less intense laminin α5 immunofluorescence staining (Figure 1B) and a significant reduction in the laminin α5-positive area (Figure 1C) compared to mock tumors. The importance of endothelial SOD3 expression on laminin α5 levels was determined using EG7-mock tumors implanted in a bi-transgenic mouse model generated by crossing the loxP-SOD3KI [37] and Tie2-Cre mouse lineages. This bi-transgenic mouse (SOD3^EC-Tg^) expresses SOD3 specifically in the tumor endothelium [32]. The laminin α5 staining intensity and stained area were reduced in tumors from these mice compared to their non-transgenic (SOD3^f/f^) counterparts (Figure 1D–F), suggesting that SOD3-induces laminin α5 downregulation.

To further investigate a potential differential regulation of *LAMA4* and *LAMA5* expression by SOD3, EG7 tumors implanted in SOD3^EC-Tg^ and SOD3^f/f^ mice were double stained with laminin α4 and laminin α5-specific antibodies. Whereas laminin α5 staining was intense in the tumors implanted in the SOD3^f/f^ mice, laminin α4 staining was markedly increased in the vessels of EG7 tumors grafted into the SOD3^EC-Tg^ mice (Figure 1G; low magnification images are provided in Appendix A). Determination of the laminin α4- and α5-stained areas in these tumors revealed a higher laminin α5/α4 ratio in the SOD3^f/f^ than in the SOD3^EC-Tg^ tumors (Figure 1H). Importantly, measurement of *LAMA4* and *LAMA5* mRNA levels revealed a reduced *LAMA5/LAMA4* ratio in tumors engrafted in SOD3^EC-Tg^ compared to those in SOD3^f/f^ mice (Figure 1I), supporting the function of SOD3 in the balance of these laminin isoforms. 

### 3.2. Extracellular SOD3 Represses LAMA5 Transcription in Endothelial Cells

The above results indicate that high SOD3 levels in the tumor milieu are associated with a reduction in laminin α5 in the endothelial BM. Intriguingly, this occurred when SOD3 was produced by tumor or endothelial cells. A mouse microvascular endothelial cell line, 1G11, modified to overexpress SOD3 [37] was used to clarify whether this laminin α5 downregulation is a direct consequence of SOD3 activity on the ECs. We found that *LAMA5* mRNA (Figure 2A) and laminin α5 protein levels as detected by immunofluorescence (Figure 2B,C) and immunoblot (Figure 2D,E and Appendix A) were significantly reduced in SOD3-overexpressing 1G11 cells. Notably, SOD3 overexpression increased laminin α4 protein levels (Figure 2D,E), as previously reported [32,33].

To rule out that the effect of SOD3 on *LAMA5* transcription was the consequence of aberrant SOD3 localization due to overexpression, *LAMA5* expression levels were analyzed in unmodified 1G11 cells incubated with extracellular SOD3. The source of this extracellular SOD3 was conditioned serum-free medium from human embryonic kidney (HEK)-293T cells transfected with murine SOD3 or hemagglutinin (HA)-tagged SOD3 (Figure 2F). Immunoblots showed the ectopic expression of both untagged and HA-tagged SOD3 in the conditioned medium (CM) (Figure 2G and Appendix A); moreover, both forms were enzymatically active (Figure 2H). In agreement with previous reports [35], extracellularly added HA-SOD3 entered into the 1G11 cells and accumulated mainly in the cytosol, although some cells showed a little nuclear staining (Figure 2I). More importantly, the addition of extracellular SOD3 significantly reduced the levels of *LAMA5* mRNA (Figure 2J) and the intensity of the laminin α5 staining (Figure 2K,L), but increased laminin α4 staining in these 1G11 cells (Appendix A), in agreement with previous reports [32]. These results indicate that extracellular SOD3 exerts a differential regulation of laminin α4 and laminin α5 in EC, which is not an artifact of overexpression.

### 3.3. LAMA5 Transcription Is Not Affected by SOD1

Experiments were next performed to determine whether other antioxidant enzymes mimic the effect of SOD3 on *LAMA5* transcription. For this, 1G11 cells stably expressing the cytosolic SOD1 isoform were generated (1G11-SOD1 cells). These expressed high SOD1 mRNA and protein levels (Figure 3A,B and Appendix A), and showed rescued superoxide levels compared to mock transfected cells, as determined by staining with the cell-permeating probe DHE after exposition with the conditioned medium from N202.1A breast cancer cells, used as a prototypic tumor cell line able to induce oxidative stress in the endothelium (Figure 3C). Moreover, SOD activity was greater in cell extracts from 1G11-SOD1 cells compared to control cells (Figure 3D). 

Tests were then undertaken to determine whether SOD1 overexpression affects *LAMA5* expression. No differences in *LAMA5* mRNA and laminin α5 protein levels were found between the 1G11-mock and SOD1-overexpressing cells (Figure 3E–G). However, both the 1G11-SOD3 and 1G11-SOD1 cells showed greater *LAMA4* mRNA transcription compared to their corresponding control cells (Figure 3H). Therefore, while SOD1 and SOD3 promote *LAMA4* expression, only SOD3 suppresses *LAMA5* expression.

### 3.4. SOD3 Modifies the Transcriptional Program of Endothelial Cells

To investigate the mechanism by which SOD3 might modulate *LAMA5* transcription, the transcriptomic profiles of 1G11-mock and 1G11-SOD3 cells were analyzed. To resemble the situation of the tumor endothelium, both subsets of ECs were exposed to the conditioned medium of N202.1A tumor cell cultures prior to analysis. Principal component analysis (PCA) clearly distinguished the 1G11-mock and 1G11-SOD3 cell gene expression profiles (Figure 4A). A total of 1682 genes showed a difference in expression (fold change [Fc] of >2 or <−2, false discovery rate [FDR] < 0.05; 1132 genes upregulated, 550 downregulated) between the mock and SOD3-overexpressing 1G11 cells (Figure 4B). These differentially expressed genes coded for cytoplasmic (40.25%), nuclear (26.13%), plasma membrane (12.93%) and extracellular (7.77%) proteins. Other differentially expressed RNAs (12.93%) represented long, non-coding RNAs, pseudogenes or unknown sequences. An 18.9 Fc increase in SOD3 expression was recorded for the 1G11-SOD3 cells, which also showed the upregulation of genes involved in the negative regulation of apoptosis (IGF-I, TNFAIP8L1, ECSCR, FAIM2), G-protein signaling (GPS1, RGS10, RGS20, RAB3B), and metabolic and mitochondrial processes (AKR1B10, NDUFC2, SOD2, MPC2, GPI1, COX6C, TIMM8A1). In contrast, SOD3 repressed genes involved in inflammation and leukocyte trafficking, including several chemokines (CCL2, CCL7, CXCL17, CXCL10) and integrins and their regulators (CIB3, VCAM-1, ICAM-1). Significant SOD3-induced *LAMA5* downregulation was also observed, whereas *LAMA4* tended to be upregulated in 1G11-SOD3 cells (Figure 4C). *WISP2*, a positive regulator of *LAMA4* [32], was also significantly upregulated (Fc + 12.36; FDR = 6.9 × 10^−5^) in these cells. 

The gene expression data were also subjected to Ingenuity Pathway Analysis (IPA). Many of the canonical pathways significantly regulated by SOD3 were related to inflammatory processes, oxidative stress and actin remodeling (Figure 4D). Among them, the Wnt/ß-catenin pathway (*p* = 0.02, z-score: 0.53), previously implicated in *LAMA4* induction in tumor endothelium [32], was activated. Four molecular networks with an IPA score of >30 (number of genes participating in a particular network) were also identified (Appendix A). The enrichment of downregulated genes involved in the “antimicrobial and inflammatory response” network is noteworthy, suggesting an anti-inflammatory activity for SOD3. In agreement with this idea, a large number of upstream transcriptional regulators related to the NF-κB pathway showed a z-score compatible with inhibition in the 1G11-SOD3 cells (Figure 4E). Indeed, NF-κB targeted the transcription of 38 differentially expressed genes (FDR < 0.05), most of them repressed in 1G11-SOD3 cells (Figure 4F). Gene set enrichment analyses (GSEA) confirmed the negative regulation of NF-κB target genes in both the canonical and non-canonical pathways in 1G11-SOD3 cells compared to mock cells (Figure 4G). Together, these data reveal the inhibition of the NF-κB pathway as a major function of SOD3 in 1G11 cells. 

### 3.5. SOD3 Attenuates the NF-κB Pathway in Endothelial Cells

To confirm the role of SOD3 affects the activation of the canonical NF-κB pathway, 1G11-mock and 1G11-SOD3 cells were stimulated with TNF-α (a well-established activator of this pathway), and the nuclear translocation of the p65 (RelA) subunit was analyzed by immunofluorescence. Under basal conditions (unstimulated), nuclear p65 levels were higher in the mock than in the SOD3-expressing cells (Figure 5A). Nonetheless, differences became significant after TNF-α stimulation, which triggered massive p65 nuclear translocation in the mock cells, but only a modest translocation in the 1G11-SOD3 cells (Figure 5A,B). This reduction of p65 nuclear translocation in the 1G11-SOD3 cells correlated with the reduced transcription of NF-κB target genes, such as *CCL2*, *CXCL10*, *FAS* and *VCAM-1* (Figure 5C). CCL2 and VCAM-1 protein levels were also reduced in 1G11-SOD3 compared to the mock cells (Appendix A), in agreement with their transcriptional downregulation. Remarkably, *ICAM-1* mRNA levels were comparable between mock- and SOD3-overexpressing cells, either in basal conditions or after TNF-α stimulation (Appendix A).

To confirm a role for SOD3 in NF-κB-mediated transcription, 1G11-mock and 1G11-SOD3 cells expressing a NF-κB-luciferase reporter construct were stimulated with conditioned medium from the N202.1A tumor cell line. This increased NF-κB promoter activity in a time-dependent manner, but more strongly in the 1G11-mock than in the 1G11-SOD3 cells (Figure 5D). 

### 3.6. SOD3 Inhibits the NF-κB Pathway by Stabilizing IκBα 

The above results support the idea that SOD3 is a negative regulator of the transcriptional activity of NF-κB. It is noteworthy that 1G11-mock and 1G11-SOD3 cells showed similar p65 protein levels (Appendix A), suggesting that SOD3 inhibits NF-κB transcription by post-translational mechanisms. The nuclear translocation of NF-κB is inhibited by IκB proteins, the degradation of which is regulated by phosphorylation [11]. To determine whether SOD3 affects IκBα degradation, 1G11-mock and 1G11-SOD3 cells were treated with cycloheximide (a protein synthesis inhibitor) and IκBα protein levels analyzed at different times after TNF-α stimulation (Figure 6A and Appendix A). IκBα levels were significantly higher in the unstimulated 1G11-SOD3 than in the mock cells; however, TNF-α stimulation reversed these differences over time (Figure 6B). Indeed, the half-life of IκBα protein was comparable between 1G11-mock (17.83 ± 6.5 min) and 1G11-SOD3 (17.97 ± 6.02 min) cells after TNF-α stimulation, suggesting that SOD3 stabilizes IκBα but does not completely prevent its degradation after a strong stimulus. Although TNF-α stimulation increased IκBα mRNA levels, no differences were seen between the 1G11-mock and 1G11-SOD3 cells (Appendix A), indicating that SOD3 regulates IκBα levels exclusively through post-translational mechanisms.

To gain further insight into how SOD3 stabilizes IκBα, IκBα ubiquitination was analyzed in unstimulated 1G11-mock and 1G11-SOD3 cells in the presence of MG132, a proteasome inhibitor. The time-dependent accumulation of phosphorylated and polyubiquitinated forms of IκBα was greater in the mock than in the SOD3-overexpressing cells (Figure 6C and Appendix A). Densitometric analyses indicated a significant reduction in phosphorylated IκBα in the 1G11-SOD3 compared to the mock cells (Figure 6D). These results indicate that SOD3 overexpression inhibits NF-κB transcription by preventing IκBα phosphorylation, a key step in IκBα ubiquitination and degradation [10].

### 3.7. SOD3 Regulates LAMA5 Expression via the NF-κB Pathway 

The NF-κB pathway has been reported to be a positive regulator of *LAMA5* in colorectal cancer cells [47]. In the present work, NF-κB pathway activation was also seen to induce *LAMA5* transcription in ECs in a time-dependent manner (Figure 7A). *LAMA4* mRNA levels were, however, not affected by TNF-α stimulation, suggesting a specific effect of NF-κB on the *LAMA5* promoter. The NF-κB-mediated regulation of *LAMA5* was confirmed by the treatment of the 1G11-mock cells with BAY11-7082, which inhibits IκBα phosphorylation and hence NF-κB activation; this reduced *LAMA5* expression and the intensity of laminin α5 staining (Figure 7B–D). Since SOD3 inhibits the NF-κB pathway, it was reasoned that SOD3-overexpressing cells should prevent *LAMA5* induction after NF-κB stimulation. Indeed, TNF-α stimulation induced *LAMA5* expression in 1G11-mock but not in 1G11-SOD3 cells (Figure 7E–G). These results suggest that high SOD3 levels reduce endothelial *LAMA5* expression via the inhibition of the NF-κB pathway. It is remarkable that SOD3 overexpression in human microvascular cells (HDMEC) downmodulated *LAMA5* mRNA levels while increased those of *LAMA4*, and this was associated with a downregulation of NF-κB target genes, such as *CCL2*, *CXCL10* and *VCAM-1*, compared to mock-transduced HDMEC cells (Appendix A). These results suggest that SOD3/NF-κB/*LAMA5* pathway is also operative in human endothelial cells.

## 4. Discussion

The balance of laminin α4/laminin α5 isoforms in the endothelial BM plays an instructive function in regulating the diapedesis of immune cells [23]. Understanding the molecular circuits that regulate the levels of these isoforms in the endothelial BM of tumor blood vessels might therefore be key in improving immune surveillance for cancer. The present work suggests that the antioxidant enzyme SOD3 is a negative regulator of laminin α5 expression in tumor ECs. SOD3 overexpression or incubation of an endothelial cell line with extracellularly added SOD3 was found to reduce *LAMA5* mRNA and laminin α5 staining in ECs. SOD3 also prevented the induction of *LAMA5* expression by NF-κB, suggesting this pathway to be a major target for SOD3-induced *LAMA5* repression. High levels of SOD3 in the milieu of EG7-SOD3 tumors grafted in syngeneic mice or EG7 tumors implanted in SOD3^EC-Tg^ mice were also correlated with a reduction in laminin α5 staining intensity in tumor blood vessels, supporting a role for SOD3/*LAMA5* interactions in vivo. Importantly, SOD3 levels were also positively associated with high laminin α4 immunostaining in these tumors, which is likely to shift the laminin α4/α5 balance towards the laminin α4high/laminin α5 low phenotype observed in postcapillaries at sites of leukocyte extravasation [26,28,30]. Indeed, SOD3 has been linked with increased T cell infiltration and immune surveillance in mouse and human tumors, an effect dependent on NO-induced stabilization of HIF-2α in ECs [33]. A scenario is therefore proposed in which high levels of SOD3 differentially regulates the transcription of *LAMA4* and *LAMA5* specifically in tumor endothelium by acting on distinct transcriptional regulators: the stabilization of HIF-2α for *LAMA4* induction, and the inhibition of NF-κB for *LAMA5* repression (see graphical abstract).

A corollary to the model proposed above is that high levels of SOD3 might increase the infiltration of anti-tumor effector T cells into the TME by reducing *LAMA5* expression in ECs; since high density of CD8^+^ T cells has been associated to better prognosis in some types of cancer, *LAMA5* levels might thus influence tumor prognosis. Supporting this hypothesis, analysis of TCGA data with TIMER algorithm showed a negative correlation between *LAMA5* mRNA levels and the intratumor CD8^+^ T cell signature for some cancers types, including stomach, lung and ovarian (Appendix A). Notably, high *LAMA5* expression in tumors negatively impacted the survival of patients with gastric, lung and ovary tumors (Appendix A). Although these data suggest an association between *LAMA5*, CD8^+^ T cell density and tumor prognosis, they must be taken with caution since *LAMA5* mRNA levels recorded in these databases are derived from the whole tumor tissue, without distinction between neoplastic and endothelial cells. Laminin-α5 can be a component of the basement membrane of the tumor epithelium, regulating tumor cell proliferation and migration [48]. Therefore, laminin-α5 might have different activities in cancer prognosis depending of its spatial compartmentalization.

In apparent contradiction, whereas SOD3 enhances tumor infiltration of effector T cells in the TME by changing the laminin α4/α5 balance, also reduces the expression of endothelial adhesion molecules required for leukocyte diapedesis. A possible explanation for this conundrum is that, in contrast to previous studies [49], we found no changes in *ICAM-1* transcription associated to SOD3 overexpression (Appendix A). High ICAM-1 expression levels on endothelial cells have been linked to T cell transcellular diapedesis [50,51], which is also the preferred pathway for diapedesis of effector and effector memory CD8+ T cells [52]. Further research is needed to clarify the role of SOD3 on ICAM-1 transcription in the tumor endothelium.

The present results are consistent with the concept that SOD3 inhibits inflammation by targeting the NF-κB pathway [49,53,54,55,56], potentially by inhibiting IκB phosphorylation. IκB proteins inhibit NF-κB translocation into the nucleus, an inhibitory mechanism overturned by IκB phosphorylation (catalyzed by the IKK complex) [14]. In the present work, SOD3 overexpression did not change IκBα mRNA levels but reduced IκBα phosphorylation and consequently its degradation, increasing IκBα levels and its inhibitory capacity. It should be noted that major differences in IκBα levels were seen between the mock-transfected and SOD3-overexpressing cells in the basal state, and that these differences disappeared upon TNF-α stimulation. It may be that IκB does not totally prevent the constitutive shuttling of NF-κB between the cytosol and nucleus, and that a low level of transcriptional activity occurs in non-activated cells [57]. This basal NF-κB activity would depend on the dissociation/re-association kinetics of the IκB/p65RelA interaction [58], which in turn is determined by the concentration of the reactants. Immunoblot analyses showed SOD3 to reduce IκBα phosphorylation and consequently increase the IκBα protein available under basal conditions, changing the IκB/p65RelA ratio towards a more inhibitory state. After strong stimulation, SOD3 attenuates but does not completely inhibit IκBα phosphorylation. Therefore, SOD3 seems not to fully prevent the ubiquitination or proteasomal degradation of IκB.

The precise mechanism by which SOD3 impairs IκBα phosphorylation deserves future exploration. A potential mediator might be NO; this is known to interfere with IκBα phosphorylation [59,60], and its levels are increased by the antioxidant activity of SOD3 [37]. However, an intriguing observation is that *LAMA5* transcription is not affected by SOD1, another antioxidant enzyme that can increase NO levels in the cytoplasm [61]. Indeed, both SOD1 and SOD3 comparably induced *LAMA4* transcription in 1G11 cells. This suggests that additional SOD3-specific mechanism(s) control the NF-κB pathway. 

The relatively large number of genes showing changes in their expression due to SOD3 overexpression was unexpected. Some of these differentially regulated genes produce products involved in processes that are likely regulated by an antioxidant enzyme, such as NO signaling, the response to oxidative stress, or the activation of signaling pathways previously reported as influenced by SOD3, such HIF and WNT/β-catenin [32,37]. Some of the other processes identified, for example, the regulation of the sumoylation pathway, were unexpected and merit further study. It is also evident that many of the predicted regulators affected by SOD3 are controllers of inflammation; in addition to NF-κB, SOD3 might influence signaling pathways elicited by Toll-like receptors, type-II interferons and STAT transcription factors. It may be that SOD3 influences all these regulators by affecting a single master regulator of inflammation, such as NF-κB. Nonetheless, the observation that SOD3 can enter the nucleus of ECs opens up the possibility that it alters transcription in these cells by modifying epigenetic regulators through redox-induced or NO-mediated reactions [62,63,64].

## 5. Conclusions

In conclusion, this study shows that SOD3 represses the transcription of *LAMA5* in tumor ECs. This activity is SOD3-specific and not shared by the anti-oxidant enzyme SOD1. SOD3-induced *LAMA5* repression is associated to the inhibition of the NF-κB pathway, through a mechanism involving the prevention of IκBα phosphorylation and consequently its degradation. These results, plus those of earlier studies [32], highlight the potential of SOD3 to be a major player in the regulation of laminin α4 and laminin α5 in the endothelial BM of tumor vasculature, shifting the laminin α4/α5 balance towards a phenotype observed in postcapillaries at sites of leukocyte extravasation. SOD3 may thus increase the infiltration of anti-tumor effector cells into the TME, explaining the better prognosis associated with high SOD3 levels in some types of cancer.

## Figures and Tables

**Figure 1 cancers-14-01226-f001:**
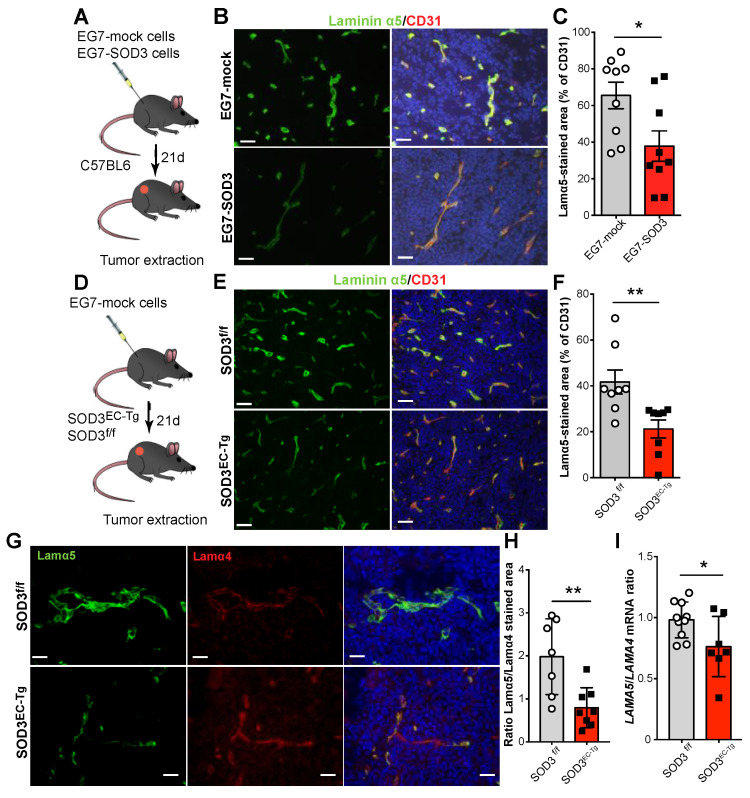
SOD3 downregulates laminin α5 and modifies the laminin α5/laminin α4 ratio in the tumor vasculature. (**A**) Diagram of tumor generation by subcutaneous (s.c.) injection of EG7-mock or EG7-SOD3 cells into C57BL6 mice (*n* = 7 mice/group). (**B**) Laminin α5 (green) and CD31 (red) staining from tumors as in **A**; nuclei were DAPI counterstained (blue). Representative images are shown. Scale bars 50 µm. (**C**) Stained laminin α5 and CD31 areas were quantified using Image J software with images as in B. The stained laminin α5 area is expressed as the percentage of that determined for CD31 (*n* = 9 fields/condition). * *p* < 0.05; two-tailed *t*-test. (**D**) Diagram of s.c. EG7 tumors in endothelial cell-specific SOD3 transgenic (SOD3^EC-Tg^) and control (SOD3^Cre-^) mice (*n* = 9 mice/group). (**E**) Representative images of laminin α5 (green) and CD31 (red) staining of EG7 tumors in SOD3^EC-Tg^ and SOD3^f/f^ mice; nuclei were DAPI counterstained (blue). Scale bars 50 µm. (**F**) Quantification of stained laminin α5 area expressed as a percentage of the stained CD31 area using images as in **E** (*n* = 8–9 fields/condition). ** *p* < 0.01; two-tailed *t*-test. (**G**) Representative images of stained laminin α5 (green) and laminin α4 (red) EG7 tumors implanted in SOD3^f/f^ and SOD3^EC-Tg^ mice; nuclei were stained with DAPI (blue). Scale bars 25 µm. (**H**) Quantification of the laminin α5/laminin α4 ratio, determined as that between the stained area for each protein (*n* = 7–8 fields/condition). ** *p* < 0.01; Mann-Whitney test. (**I**) Ratio of the relative *LAMA5* and *LAMA4* mRNA levels isolated from EG7 tumors grafted into SOD3^Cre-^ and SOD3^EC-Tg^ mice. Each dot represents the mean of a triplicate from an independent tumor sample. * *p* < 0.05; Mann-Whitney test. The tumor growth kinetics, survival data and SOD3 expression of the tumors here analyzed have been previously reported [32].

**Figure 2 cancers-14-01226-f002:**
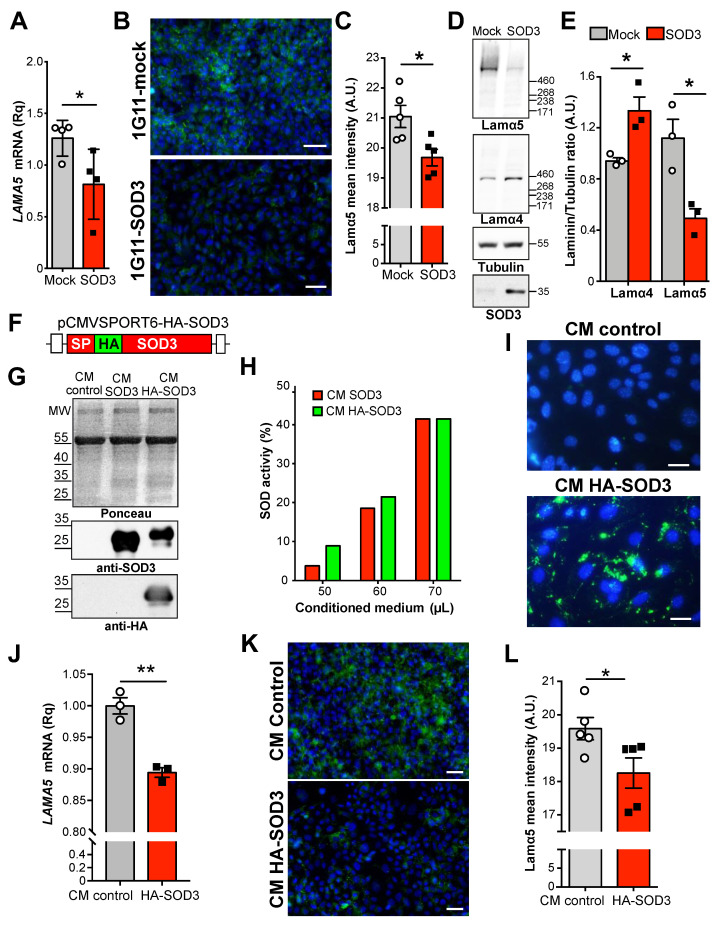
SOD3 downregulates laminin α5 expression in endothelial cells. (**A**) *LAMA5* mRNA levels in 1G11-mock and 1G11-SOD3 cells determined by RT-qPCR. Each dot represents the mean of a triplicate from an independent experiment. * *p* < 0.05; two-tailed *t*-test. (**B**) Laminin α5 (green; antibody 504) staining in 1G11-mock and 1G11-SOD3 cells; nuclei were DAPI counterstained (blue). Representative images are shown. Scale bars 50 µm. (**C**) Quantification of laminin α5 mean fluorescence intensity from images as in **B** (*n* = 5 fields/condition). * *p* < 0.05; two-tailed *t*-test. (**D**) Cell extracts from 1G11-mock and 1G11-SOD3 were resolved by 4–12% SDS-PAGE and sequentially immunoblotted with anti-laminin α5, anti-laminin α4, anti-SOD3 and anti-α tubulin, as loading control. (**E**) Densitometric quantification of laminin α5/tubulin and anti-laminin α4/tubulin ratio from immunoblots as in (**D**) (*n* = 3). * *p* < 0.05; two-tailed *t*-test. (**F**) Diagram of the HA-tagged SOD3 form cloned in the pCMV-SPORT6 vector. SP, signal peptide; HA, influenza hemagglutinin. (**G**) Conditioned medium (CM) from cells transfected with the empty vector (control), SOD3 or HA-SOD3, resolved by SDS-PAGE. Transferred proteins were visualized with Ponceau red (**upper panel**) and then immunoblotted with anti-SOD3 (**middle**) and anti-HA antibodies (**lower**). Representative of three experiments. (**H**) SOD activity in SOD3 and HA-SOD3 CM. (**I**) 1G11 cells were incubated with control or HA-SOD3 CM and SOD3 visualized with an anti-HA antibody (green); nuclei were DAPI-counterstained (blue). Scale bars 25 µm. (**J**) Relative *LAMA5* mRNA levels in 1G11 cells incubated with control or HA-SOD3 CM (*n* = 3). ** *p* < 0.01; two-tailed *t*-test. (**K**) Laminin α5 staining (green; antibody 504) in 1G11 cells incubated with control or HA-SOD3 CM; nuclei were DAPI-counterstained (blue). Scale bars 50 µm. (**L**) Quantification of laminin α5 mean fluorescence intensity from (**K***)* (*n* = 5 fields/condition). * *p* < 0.05; two-tailed *t*-test.

**Figure 3 cancers-14-01226-f003:**
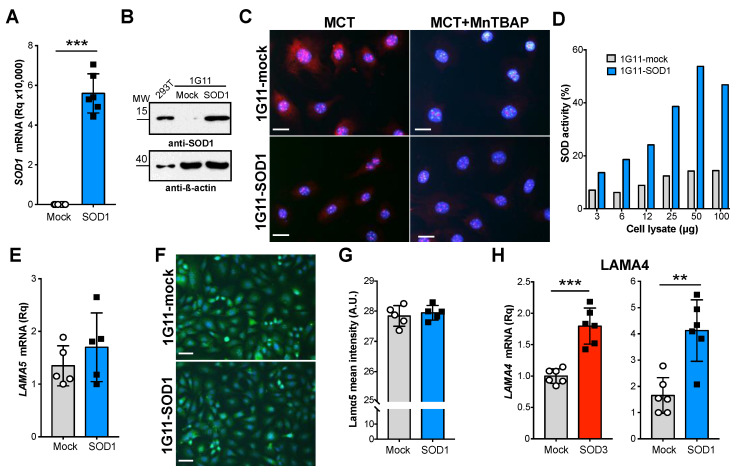
SOD1 does not affect laminin α5 expression in endothelial cells. (**A**) mRNA levels of human SOD1 in 1G11-mock and 1G11-SOD1 cells, as determined by RT-qPCR. Each dot represents the mean of a triplicate from an independent experiment *** *p* < 0.001; two-tailed *t*-test. (**B**) Human SOD1 protein levels in 1G11-mock and 1G11-SOD1 cells as determined by immunoblotting; HEK-293T cells were used as a human SOD1 reference. Immunoblots were rehybridized with an anti-β-actin antibody as a loading control. A representative experiment is shown. (**C**) 1G11-mock and 1G11-SOD1 cells were incubated (14 h) with the conditioned medium from N202.1A tumor cell line cultures (MCT) and then stained with DHE (red); nuclei were DAPI-counterstained (blue). Treatment with the SOD mimetic MnTBAP was used as control (right panels). Scale bars 25 µm. (**D**) SOD activity in 1G11-mock and 1G11-SOD1 cell extracts. (**E**) Relative *LAMA5* mRNA levels in 1G11-mock and 1G11-SOD1 cells. Each dot represents the mean of a triplicate from an independent experiment (*n* = 5). *p* = 0.32, two-tailed *t*-test. (**F**) Representative images of laminin α5 (green; PAC078MV01) staining in 1G11-mock and 1G11-SOD1 cells; nuclei were DAPI counterstained (blue). Scale bars 50 µm. (**G**) Quantification of laminin α5 mean fluorescence intensity from the images as in (**F**) (*n* = 5 fields/condition). *p* = 0.58; two-tailed *t*-test. (**H**) Relative *LAMA4* mRNA levels in 1G11-mock, 1G11-SOD3 and 1G11-SOD1 cells. Each dot represents the mean of a triplicate from an independent experiment (*n* = 6). ** *p* < 0.01, *** *p* < 0.001, two-tailed *t*-test.

**Figure 4 cancers-14-01226-f004:**
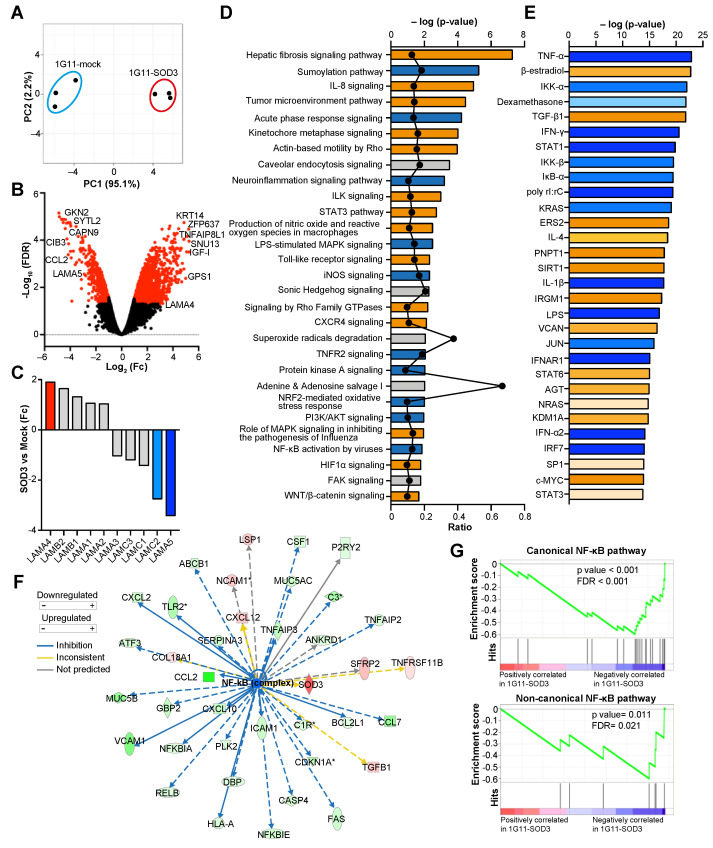
SOD3 induces a specific transcriptional program in 1G11 endothelial cells. (**A**) PCA plot using the rLog-transformed values from the microarray analysis of 1G11-mock and 1G11-SOD3 cells. Each dot represents an independent experiment. (**B**) Volcano plot of differentially expressed genes in 1G11-mock and 1G11-SOD3 cells. Red dots indicate genes with an absolute fold change (Fc) > 2 and an FDR of <0.05. Some of the genes with the highest variation are indicated. (**C**) Average Fc of the laminin chains as detected by the microarray (*n* = 3). (**D**) Selection of Ingenuity Pathway Analysis (IPA) top canonical pathways regulated by SOD3, listed according to their *p* value (lowest *p* = 0.02). The color of the bars indicates pathway activation (orange) or inhibition (blue) based on a Z-score; gray indicates no activity pattern available. The black circles represent the ratio of the number of differentially expressed genes in each pathway over the total number of genes in that pathway. (**E**) Top 30 upstream regulators identified by IPA (lowest *p*-value = 1.7 × 10^−14^). Colors indicate activation (orange) or inhibition (blue) based on a statistical Z-score. (**F**) Upstream regulator analysis of differentially expressed genes in 1G11-SOD3 cells related to the NF-κB network. Color code is indicated at the left. (**G**) GSEA of the canonical (**top**) and non-canonical (**bottom**) NF-κB target genes in 1G11-SOD3 and 1G11-mock cells. Enrichment plots (green); the ranked list metric for each gene as a function of the rank in the ordered dataset (bottom) is shown. The *p* values and FDRs are indicated.

**Figure 5 cancers-14-01226-f005:**
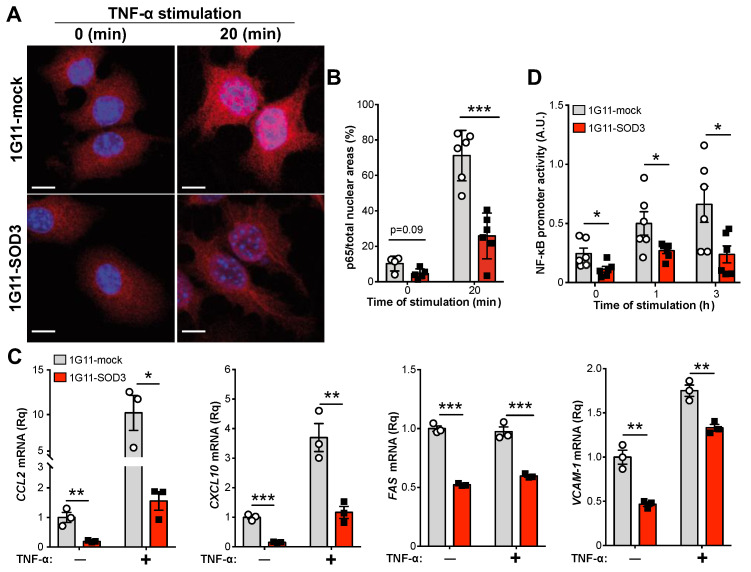
SOD3 attenuates NF-κB transcriptional activity in endothelial cells. (**A**) Representative images of RelA (p65) staining (red) in unstimulated (0) and TNF-α-stimulated (20 min) 1G11-mock and 1G11-SOD3 cells; nuclei were DAPI-counterstained (blue). Scale bars 10 µm. (**B**) Percentage of the nuclear p65 stained area with reference to the total nuclear area as determined by DAPI staining (*n* = 4–6 fields/condition). *** *p* < 0.001, two-way ANOVA with Bonferroni post-hoc correction. (**C**) Relative CCL2, CXCL10, FAS and VCAM-1 mRNA levels in unstimulated and TNF-α-stimulated (1.5 h for CCL2 and CXCL10; 0.5 h for FAS and VCAM-1) 1G11-mock and 1G11-SOD3 cells (*n* = 3). Each dot represents the mean of a triplicate from an independent experiment. * *p* < 0.05, ** *p* < 0.01, *** *p* < 0.001, two-way ANOVA with Bonferroni post-hoc correction. (**D**) 1G11-mock and 1G11-SOD3 cells were transiently transfected with the NF-κB-luciferase firefly reporter and the promoter-less pRL-SV40-luciferase *Renilla* plasmids, and then stimulated with conditioned medium from N202.1A tumor cells. The relative NF-κB promoter activity was determined by the firefly/*Renilla* ratio (*n* = 6). * *p* < 0.05, two-tailed *t*-test.

**Figure 6 cancers-14-01226-f006:**
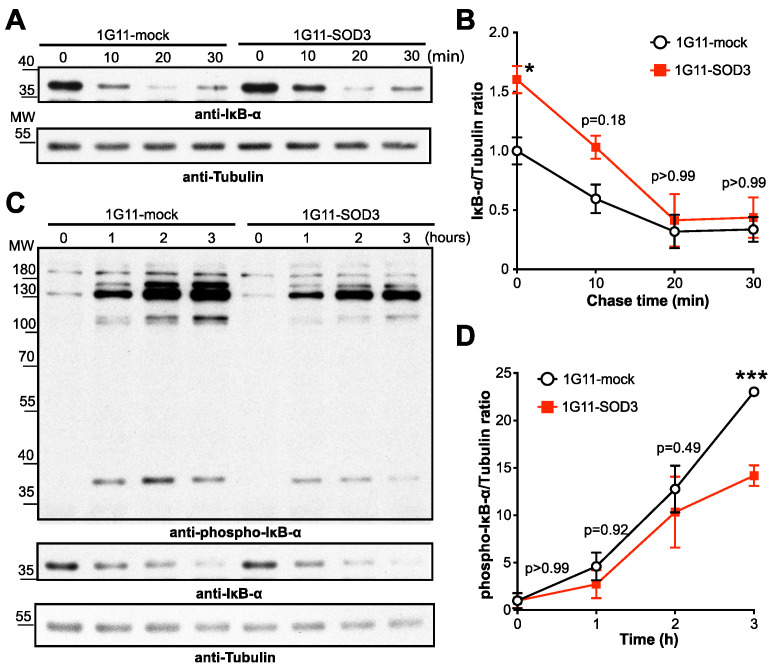
SOD3 attenuates IκBα phosphorylation and degradation. (**A**) 1G11-mock and 1G11-SOD3 cells were treated with cycloheximide and then stimulated with TNF-α for the indicated times (unstimulated cells as 0). Representative immunoblots with anti-IκBα and anti-tubulin (loading control) antibodies are shown. (**B**) Immunoblots as in A were quantified by densitometry and the IκBα/tubulin ratio determined (*n* = 3). * *p* < 0.05, two-way ANOVA with Bonferroni post-hoc correction. (**C**) Equal numbers of 1G11-mock and 1G11-SOD3 cells were treated with MG-132. Cell lysates were obtained at the indicated times and then sequentially immunoblotted with anti-phospho IκBα, anti-IκBα and anti-tubulin (loading control) antibodies. (**D**) The phospho IκBα/tubulin ratio was determined by densitometric analysis of the immunoblots in (**C**) (*n* = 3). *** *p* < 0.001, two-way ANOVA with Bonferroni post-hoc correction.

**Figure 7 cancers-14-01226-f007:**
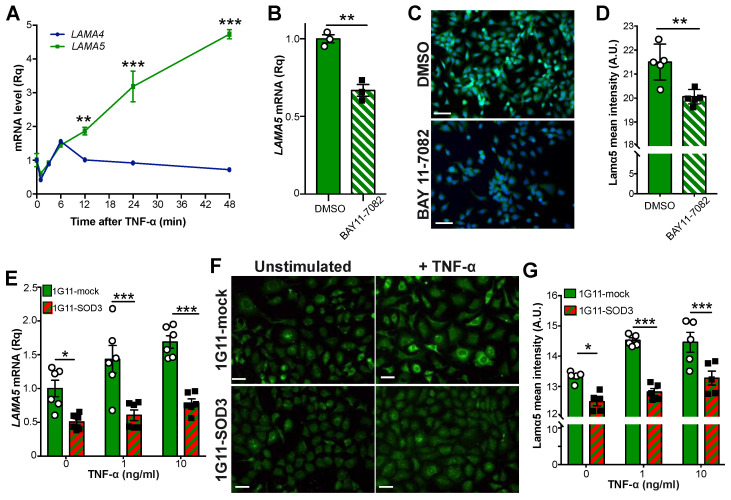
SOD3 restrains NF-κB-induced LAMA5 expression. (**A**) Time-course changes in the relative *LAMA4* and *LAMA5* mRNA levels in 1G11-mock and 1G11-SOD3 cells after TNF-α stimulation. Values were normalized against the unstimulated condition (time 0) (*n* = 3 independent experiments). ** *p* < 0.01, *** *p* < 0.001, two-way ANOVA with Bonferroni post-hoc correction. (**B**) Relative *LAMA5* mRNA levels in vehicle- (DMSO) or BAY11-7082-treated 1G11 cells. ** *p* < 0.01, two-tailed *t*-test. (**C**,**D**) Representative images and quantification of laminin α5 (green; PAC078MV01) mean fluorescence intensity in vehicle- (DMSO) or BAY11-7082-treated 1G11 cells; nuclei were DAPI counterstained (blue) (*n* = 5 fields/condition). Scale bars 50 µm. ** *p* < 0.01, two-tailed *t*-test. (**E**) Dose-dependent induction of *LAMA5* transcription in 1G11-mock and 1G11-SOD3 cells by TNF-α. Values were normalized to unstimulated (0) cells (*n* = 6). * *p* < 0.05, *** *p* < 0.001, two-way ANOVA with Bonferroni post-hoc correction. (**F**) Representative images of laminin α5 staining (green; antibody 504) in control and TNF-α stimulated 1G11-mock and 1G11-SOD3 cells. Scale bars 50 µm. (**G**) Quantification of laminin α5 mean fluorescence intensity in cells as in (**F***)* (*n* = 5 fields/condition). * *p* < 0.05, *** *p* < 0.001, two-way ANOVA with Bonferroni post-hoc correction.

## Data Availability

Data files and normalized values from transcriptomic analyses were deposited in the NCBI Gene Expression Omnibus database (http://www.ncbi.nlm.nih.gov/geo, last accessed on 25 February 2022), and are accessible through the GEO Series accession number GSE189247. Other datasets from this study are available upon reasonable request to the corresponding authors.

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
