# Peer review of "Superoxide Dismutase-3 Downregulates Laminin α5 Expression in Tumor Endothelial Cells via the Inhibition of Nuclear Factor Kappa B Signaling"

_cancers, 2022, doi:10.3390/cancers14051226_

Round 1

Reviewer 1 Report

The authors identify SOD3 as a negative regulator of the Laminin5 gene through the NF-KappaB signalling.

The study is convincing and the conclusions are in general supported by the experimental data.

I have one comment that I would like the authors to address:

The authors use mouse EC throughout the paper. Would it be possible to test whether this same signalling axis is conserved also in human EC?

Author Response

The authors identify SOD3 as a negative regulator of the Laminin5 gene through the NF-KappaB signalling.The study is convincing and the conclusions are in general supported by the experimental data.

 We thank the reviewer for his/her comments

I have one comment that I would like the authors to address: The authors use mouse EC throughout the paper. Would it be possible to test whether this same signalling axis is conserved also in human EC?

We thank the reviewer for calling our attention to this important point. We have addressed the concern by overexpressing human SOD3 in human dermal microvascular endothelial cells (HDMEC) using an adenoviral vector. We found that SOD3 overexpression downregulated LAMA5 mRNA levels whereas increased LAMA4 transcription, compared to mock-transduced cells. In addition, SOD3 overexpression downmodulated NF-κB target genes, such as CCL2, CXCL10 and VCAM-1. Our interpretation is that these data, included in a new supplementary figure (Figure S11), support that SOD3/NF-κB/LAMA5 pathway is also operative in human endothelial cells.

Reviewer 2 Report

The authors published previous work supporting the present study. Therefore, they proposed to further characterize SOD3’s role in the tumor microenvironment, most precisely in the vascular ECM component. Focusing on the endothelial BM laminins α5/α4 deposition dynamics, the authors characterize an important effect of SOD3, overexpressed in neoplastic and endothelial cells, on this dynamics. SOD3 impaired laminin α5 expression at both transcriptional and protein levels. The mechanism responsible for the observed effect has been attributed to the inhibition of NF-κB activity. This work helps explaining previous observations relating to the improved cancer immune surveillance associated to high SOD3 levels, which is likely due to a laminin-α4high/laminin-α5low phenotype. This has been thoroughly demonstrated by both in vivo and in vitro experiments. Although the study is overall sound and well designed and the authors reached the proposed goals, a few aspects need clarification.

Major points:

Line 235 to 245 – Please add the references for the used fluorescence quantification methods;

Line 417 – Why using mammary cancer cell line and not thymoma cells for this experiment? This should be explained in the main text;

Figure 6 (C) – Do you have an explanation for why so many bands appear in the anti-phospho IκBα blot?

Discussion – Has a correlation between not only SOD3 but also laminin α5 expression levels and the prognosis of thymoma patients been described? Or any other cancer type? This analysis could be important to undestand if this mechanism improves the outcome of cancer patients, as it supposedly increases cytotoxic T cell infiltration. Analysis of these variants in databases such as The Cancer Genome Atlas (TCGA) database would improve the present work;

Line 603 – “SOD3 overexpression did not change IκBα mRNA levels but reduced IκBα phosphorylation and consequently its degradation, increasing IκBα levels…”. As you discuss next, I don’t think there is enough evidence in this work to say IκBα levels are increased, at best the results hint at a possible increase in IκBα levels, but this cannot be actually proved with this set of experiments;

Minor points:

Line 17 – Should be “… (SOD) 3...”;

Line 18 –  with enhanced disease-free survival of colorectal cancer patients. Enhanced is not usually applied in this context, improved disease-free survival is more correct;

Line 20 – “The results showed that SOD3 antagonistically regulates…”. The word antagonistically is confusing in this context, differentially regulates could be better;

Line 22 – NF-κB appears for the first time, should be written in full;

Line 39 – Would add: In conclusion, “SOD3 is a major…”;

Line 52 – “ not only …, but also use this…”;

Line 53 – replace “reversion” with conversion;

Line 60 – “TME” should be between parentheses;

Line 61 – “not only…, but also greatly increases tumor…”;

Line 83 – same comment as lines 52 and 61;

Line 137 – “…and EG7-SOD3 mouse thymoma cell lines…”;

Line 138 – “… murine mammary cancer cells…”;

Line 143 – All commercial brands should include the city and country. Please add throughout the text when first mentioned;

Line 158 – How many animals were used?

Line 219 – Immunofluorescene staining;

Line 225 – “anti-hemagglutinin (HA) tag antibody”;

Line 230 – anti-laminin α5 (antibody 504) is it also from santa cruz? If not please add info on the origin of this antibody;

Line 252 – “SOD3 (M-106, sc-67089) (Santa Cruz Biotechnology) or anti-HA…”;

Line 253 – “antibodies, respectively.”;

Line 260 – please write CHX in full;

Line 263 – please add reference for densitometry calculation using ImageJ software;

Line 271 – Write DHE in full

Line 296 – please add company for prism software;

Line 308 – CD31+ staining is supposed to be a endothelial cell maker, correct? If so might be go to mention it.

Figure 1 – please add the N of the included animals in (A) and (D) schemes;

Line 373 – again differential might be clearer than antagonistic;

Line 377 – “Each dot represents the mean of a triplicate from an independent tumor sample”. This phrase is not clear to me, in figure 2 the experiments were not performed in vitro in a mouse microvascular endothelial cell line? Or was this cell line isolated from the tumors from figure 1? This needs clarification.

Line 393 – Capitalize “Laminin” in the beginning of the sentence;

Line 395 – please represent (K) between parentheses;

Line 403 – please replace “dihydroethidium” for DHE, for it was introduced in the methods section;

Line 407 – (Fig. 3E-G);

Figure 3 – State in the legend what MCT in (C) means;

Line 412 – “SOD1 does not affect laminin α5 expression in endothelial cells.” Should be in italic, like the other figure legends.

Line 413 – same comment as for line 377;

Line 415 – “human HEK-293T cells were used as a human SOD1 reference”: this information should also be included in the main text, for example in the method section when the HEK cells are mentioned first;

Line 418 – the compound MnTBAP appears for the first time here, it should have been introduced in the method section;

Line 420 – same comment as for line 377;

Line 423 – please represent (F) between parentheses;

Line 424 – same comment as for line 377;

Line 454 – “Accumulation” might not be the best word to clearly convey that information;

Line 481 – In (C) the SD/SEM bars are not included in the graph although 3 independent experiments were performed. Is this a representative graph of 1 experiment? Otherwise SD/SEM bars should be added;

Line 492 – “To confirm the role of…”;

Line 499 – Instead of “lessens”, attenuates could be an alternative;

Line 506 – “independent sample” or experiment?

Line 515 – not shown p65 protein levels could be included in the suplemmentary data;

Line 518 – IκBα should be written in full the first time it appears in the text;

Line 528 – same comment as for line 515;

Line 539 – please represent (C) between parentheses;

Line 575 – please represent (F) between parentheses;

Line 587 – state the type of tumor of the mouse model;

Line 589 – “in vivo” should be in italic;

Line 595 – same comment as for lines 20 and 373;

Line 601 – either IκB proteins inhibit or IκB protein inhibits;

Line 623 – “suggests that additional”;

Line 650 – same comment as for lines 20 and 373;

These comments should be applied also to the supplementary data legends.

Author Response

The authors published previous work supporting the present study. Therefore, they proposed to further characterize SOD3’s role in the tumor microenvironment, most precisely in the vascular ECM component. Focusing on the endothelial BM laminins α5/α4 deposition dynamics, the authors characterize an important effect of SOD3, overexpressed in neoplastic and endothelial cells, on this dynamics. SOD3 impaired laminin α5 expression at both transcriptional and protein levels. The mechanism responsible for the observed effect has been attributed to the inhibition of NF-κB activity. This work helps explaining previous observations relating to the improved cancer immune surveillance associated to high SOD3 levels, which is likely due to a laminin-α4high/laminin-α5low phenotype. This has been thoroughly demonstrated by both in vivo and in vitro experiments. Although the study is overall sound and well designed and the authors reached the proposed goals, a few aspects need clarification.

 We thank the reviewer for his/her comments

  • Major points:
  • Line 235 to 245 – Please add the references for the used fluorescence quantification methods;

We do not understand this reviewer’s comment. We describe the method used for fluorescence quantification, which basically coincides with that we and others have described in other studies for quantification of images using the Image J software. We think that this thorough explanation of the method is more useful for readers than a simple reference to another manuscript. Nonetheless, we will be delighted to discuss with the reviewer any particular concern about the method used.

  • Line 417 – Why using mammary cancer cell line and not thymoma cells for this experiment? This should be explained in the main text;

We consider that any cancer cell line would secrete factors inducing oxidative stress in endothelial cells (due to the secretion of inflammatory mediators, etc). We used conditioned medium from N2021-A cell line because it was available and previously tested in the laboratory. This has been clarified in the text.

  • Figure 6 (C) – Do you have an explanation for why so many bands appear in the anti-phospho IκBα blot?

The high molecular weight bands correspond to polyubiquitinated forms of phospho-IκBα. It should be noted that the proteasome is inhibited in this experiment and consequently, polyubiquitinated forms are not degraded. This is indicated in the text.

  • Discussion – Has a correlation between not only SOD3 but also laminin α5 expression levels and the prognosis of thymoma patients been described? Or any other cancer type? This analysis could be important to undestand if this mechanism improves the outcome of cancer patients, as it supposedly increases cytotoxic T cell infiltration. Analysis of these variants in databases such as The Cancer Genome Atlas (TCGA) database would improve the present work;

Our work suggests that laminin-α5 levels in the endothelial basal membrane might determine CD8+ T cell extravasation into tumors. Given the association between CD8+ T cell infiltration and the prognosis in some cancers, endothelial laminin-α5 could be negatively associated with clinical evolution of the patients. Unfortunately, transcriptomic data from the TCGA Consortium and other public databases are probably not the best source to validate this hypothesis, since they are derived from the whole tumor tissue (i.e., without discrimination of neoplastic and stromal cells). Furthermore, laminin-α5 is also a component of the basement membrane of the tumor epithelium, and it can enhance tumor cell proliferation and migration (see for instance, Pouliot and Kusuma. 2013. https://doi.org/10.4161/cam.22125). Therefore, laminin-α5 might have different  activities depending on its spatial compartmentalization.

Despite these considerations, we have tried to approach the reviewer’s question by performing two type of analyses. On the one hand, we used the algorithm TIMER to determine the correlation between LAMA5 expression level and the infiltration of CD8+ T lymphocytes according to the TCGA databases. This analysis showed a negative correlation between LAMA5 and CD8+ T cell signature in nine tumor types, including thymoma, stomach, and lung and ovarian cancers, among others. On the other hand, we used the Kaplan-Meier Plotter algorithm to determine the association of LAMA5 levels with progression-free survival (PFS) or the overall survival (OS) in selected cancer types. We found a significant positive association between low LAMA5 expression and increased progression-free survival (PFS) or overall survival (OS) for gastric, ovarian and lung cancers. In the case of thymoma (data from the TCGA database), the survival rate was very high (~>80%) in the two groups (high and low LAMA5) of patients, which precluded to raise conclusions. Finally, for breast cancer we also found a significant association between low laminin-α5 protein levels and OS. This information has been included in the new supplementary Table S2 and FigureS7 and discussed in the manuscript, although we consider that they must be taken with extreme caution given the arguments provided above.

  • Line 603 – “SOD3 overexpression did not change IκBα mRNA levels but reduced IκBα phosphorylation and consequently its degradation, increasing IκBα levels…”. As you discuss next, I don’t think there is enough evidence in this work to say IκBα levels are increased, at best the results hint at a possible increase in IκBα levels, but this cannot be actually proved with this set of experiments;

We thank the reviewer to call our attention on this important point. The sentence has been modified to eliminate the “increasing IκB-α levels”

  • Minor points:

We thank the reviewer for this detailed analysis of the manuscript. When nothing is indicated, we have corrected the text according to reviewer’s suggestions. In other cases, an explanation/commentary has been added.

  • Line 17 – Should be “… (SOD) 3...”;
  • Line 18 –  with enhanced disease-free survival of colorectal cancer patients. Enhanced is not usually applied in this context, improved disease-free survival is more correct;
  • Line 20 – “The results showed that SOD3 antagonistically regulates…”. The word antagonistically is confusing in this context, differentially regulates could be better;
  • Line 22 – NF-κB appears for the first time, should be written in full;
  • Line 39 – Would add: In conclusion, “SOD3 is a major…”;
  • Line 52 – “ not only …, but also use this…”;
  • Line 53 – replace “reversion” with conversion;
  • Line 60 – “TME” should be between parentheses;
  • Line 61 – “not only…, but also greatly increases tumor…”;
  • Line 83 – same comment as lines 52 and 61;
  • Line 137 – “…and EG7-SOD3 mouse thymoma cell lines…”;
  • Line 138 – “… murine mammary cancer cells…”;
  • Line 143 – All commercial brands should include the city and country. Please add throughout the text when first mentioned;
  • Line 158 – How many animals were used?
  • Line 219 – Immunofluorescene staining;
  • Line 225 – “anti-hemagglutinin (HA) tag antibody”;
  • Line 230 – anti-laminin α5 (antibody 504) is it also from santa cruz? If not please add info on the origin of this antibody;

This antibody is produced by one the coauthors and has been described in ref. 45. This is indicated at Line 236.

  • Line 252 – “SOD3 (M-106, sc-67089) (Santa Cruz Biotechnology) or anti-HA…”;
  • Line 253 – “antibodies, respectively.”;
  • Line 260 – please write CHX in full;
  • Line 263 – please add reference for densitometry calculation using ImageJ software;

We do not understand this referee’s question. There is only one way to calculate densitometry…

  • Line 271 – Write DHE in full
  • Line 296 – please add company for prism software;
  • Line 308 – CD31+ staining is supposed to be a endothelial cell maker, correct? If so might be go to mention it.
  • Figure 1 – please add the N of the included animals in (A) and (D) schemes;
  • Line 373 – again differential might be clearer than antagonistic;
  • Line 377 – “Each dot represents the mean of a triplicate from an independent tumor sample”. This phrase is not clear to me, in figure 2 the experiments were not performed in vitro in a mouse microvascular endothelial cell line? Or was this cell line isolated from the tumors from figure 1? This needs clarification.

Our apologies; it was a “cut/paste” mistake. Now it is clearly stated that these experiments involved 1G11 cells. This point has been also corrected in other figure legends.

  • Line 393 – Capitalize “Laminin” in the beginning of the sentence;
  • Line 395 – please represent (K) between parentheses;
  • Line 403 – please replace “dihydroethidium” for DHE, for it was introduced in the methods section;
  • Line 407 – (Fig. 3E-G);
  • Figure 3 – State in the legend what MCT in (C) means;
  • Line 412 – “SOD1 does not affect laminin α5 expression in endothelial cells.” Should be in italic, like the other figure legends.
  • Line 413 – same comment as for line 377;
  • Line 415 – “human HEK-293T cells were used as a human SOD1 reference”: this information should also be included in the main text, for example in the method section when the HEK cells are mentioned first;

This was already indicated in the Immunoblotting section. The sentence has been modified to make it clearer.

  • Line 418 – the compound MnTBAP appears for the first time here, it should have been introduced in the method section;

This compound was mentioned in the “SOD activity assays” section. We have now included the whole name of the compound.

  • Line 420 – same comment as for line 377;
  • Line 423 – please represent (F) between parentheses;
  • Line 424 – same comment as for line 377;
  • Line 454 – “Accumulation” might not be the best word to clearly convey that information; “Accumulation” has been substituted by “Enrichment”
  • Line 481 – In (C) the SD/SEM bars are not included in the graph although 3 independent experiments were performed. Is this a representative graph of 1 experiment? Otherwise SD/SEM bars should be added;

Each bar represents a quotient calculated from the average expression of the indicated genes in 1G11-mock and -SOD3 cells in the array. Therefore it is not possible to get SD/SEM. These average expression data were the same as used for Volcano plots and derived from FIESTA analysis of the microarray dataset.

  • Line 492 – “To confirm the role of…”;
  • Line 499 – Instead of “lessens”, attenuates could be an alternative;
  • Line 506 – “independent sample” or experiment?
  • Line 515 – not shown p65 protein levels could be included in the suplemmentary data;
  • Line 518 – IκBα should be written in full the first time it appears in the text;

Following reviewer’s recommendation we have included the full name of IκB-α in the abstract (the first time it appears). As a consequence of this change, the abstract wordcount now exceed the 200 words limit.

  • Line 528 – same comment as for line 515;
  • Line 539 – please represent (C) between parentheses;
  • Line 575 – please represent (F) between parentheses;
  • Line 587 – state the type of tumor of the mouse model;
  • Line 589 – “in vivo” should be in italic;
  • Line 595 – same comment as for lines 20 and 373;
  • Line 601 – either IκB proteins inhibit or IκB protein inhibits;
  • Line 623 – “suggests that additional”;
  • Line 650 – same comment as for lines 20 and 373;
  • These comments should be applied also to the supplementary data legends.

Reviewer 3 Report

In this study the authors observed that SOD3 overexpression reduces laminin 5 expression by inhibiting NF-KB. This follows a previous study by the same authors that demonstrated that SOD3 increases tumor T-cell infiltration by in part upregulating laminin4. Overall the manuscript is clear and experiments are well controlled. Here are minor points that should be addressed:

  • The authors also demonstrates that SOD3 overexpression reduces adhesion molecules such as ICAM-1, VCAM-1 that are necessary for tissue infiltration by leucocytes. Hence SOD3 displays contradicting effects on leucocytes extravasation by one one hand favouring it via reduced laminin5 expression but on the other hand repressing it by downregulating adhesion molecules. This point should be better discussed.
  • Did the author notice any changes in co-stimulatory or inhibitory molecules such as ICOS-L, PD-L1 or PD-L2 in the transcriptomic analysis
  • Is there any therapeutic perspective? How could SOD3 expression been enhanced in tumors?

Author Response

In this study the authors observed that SOD3 overexpression reduces laminin 5 expression by inhibiting NF-KB. This follows a previous study by the same authors that demonstrated that SOD3 increases tumor T-cell infiltration by in part upregulating laminin4. Overall the manuscript is clear and experiments are well controlled. Here are minor points that should be addressed:

 We thank the reviewer for his/her comments

  • The authors also demonstrates that SOD3 overexpression reduces adhesion molecules such as ICAM-1, VCAM-1 that are necessary for tissue infiltration by leucocytes. Hence SOD3 displays contradicting effects on leucocytes extravasation by one one hand favouring it via reduced laminin5 expression but on the other hand repressing it by downregulating adhesion molecules. This point should be better discussed.

We concur with the reviewer that this point is a bit contradictory. Nevertheless, we would like to clarify that although microarray data indicated a slight decrease of ICAM-1 expression in 1G11-SOD3 compared to mock, RT-qPCR analyses did not confirmed this downregulation, in neither basal or TNF-a-stimulated cells. Analysis of ICAM-1 mRNA levels by RT-qPCR has been included in Figure S4C. With this data in hand, is that SOD3 only downregulates VCAM-1 but not ICAM-1 levels. It is important to highlight that ICAM-1 is a major regulator of transcellular migration diapedesis for T cells in different settings and, interestingly, a recent report has shown that effector and effector memory T cells cross the endothelium predominantly in a transcellular fashion. Therefore, an attractive hypothesis could be that SOD3 does transcellular diapedesis of effector T cells but makes endothelial BM more permissive acting of LAMA4/LAMA5 transcription, with a net  enhancement of TILs in the TME. A brief discussion has been including in the text.

  • Did the author notice any changes in co-stimulatory or inhibitory molecules such as ICOS-L, PD-L1 or PD-L2 in the transcriptomic analysis

We did not detect the expression of the indicated genes in mock or SOD3-expressing 1G11 cells, according to transcriptomic data.

  • Is there any therapeutic perspective? How could SOD3 expression been enhanced in tumors?

We previously reported that some statins enhance the expression of SOD3 in mouse tumors (Mira et al. 2018). Nevertheless, statins are pleiotropic drugs with potent anti-inflammatory and immunomodulatory effects in some settings. We are therefore not encouraged to propose the use of statins from a therapeutic perspective. We need to explore other pharmacological alternatives to increase SOD3 expression in the endothelium without the potential side-effects associated to statins.

Round 2

Reviewer 1 Report

The authors have addressed my question, therefore I recommend this study for pubblication